# Partial Likelihood Thompson Sampling

**Han Wu**[1]                    **Stefan Wager**[2]

[1]Department of Statistics, Stanford University
[2]Graduate School of Business, Stanford University

## Abstract

We consider the problem of deciding how best to target and prioritize existing vaccines that may offer protection against new variants of an infectious disease. Sequential experiments are a promising approach; however, challenges due to delayed feedback and the overall ebb and flow of disease prevalence make available methods inapplicable for this task. We present a method, partial likelihood Thompson sampling, that can handle these challenges. Our method involves running Thompson sampling with belief updates determined by partial likelihood each time we observe an event. To test our approach, we ran a semi-synthetic experiment based on 200 days of COVID-19 infection data in the US.

## 1 INTRODUCTION

Methods for sequential experimentation have proven themselves as powerful and versatile in a number of application areas, ranging from online advertising [Chapelle and Li, 2011] and revenue management [Ferreira et al., 2018] to website optimization [Letham et al., 2019]. These methods enable us to efficiently optimize an explore-exploit tradeoff between first discovering which of a number of actions is best and then efficiently deploying it once we've identified it. One simple yet successful idea for doing so is Thompson sampling [Thompson, 1933, Russo et al., 2018], where an agent dynamically updates a belief distribution for the probability that each action they could take is best, and then takes actions with propensity proportional to these beliefs.

An important and potentially promising application for sequential learning[1] is in deciding how best to use existing vaccines to target new variants of an infectious disease. For example, in the case of the COVID-19 pandemic, a number of vaccines were developed and found to be safe and effective in protecting against the original viral strain; however, new coronavirus variants then emerged that exhibited at least partial ability to evade protection from vaccines, thus making it more difficult to contain the pandemic [Kustin et al., 2021]. In situations like this, it is of considerable value to promptly assess which of the existing vaccines (if any) offer protection against the new variant. Castillo et al. [2021] call for embedding vaccine trials on new COVID-19 variants within national vaccine rollouts, and sequential learning seems like a perfect candidate for optimizing the resulting explore-exploit tradeoff across vaccines. In this setup, we would only be comparing vaccines that have already been established as generally safe and effective, and so the goal is to discover—as quickly as possible—which vaccine is most effective in the context of interest.

The main difficulty in using sequential experiments for vaccine trials is that such trials involve delayed feedback of a type that cannot readily be handled by available methods, including Thompson sampling [Thompson, 1933, Russo et al., 2018] or the UCB algorithm [Lai and Robbins, 1985, Auer et al., 2004]. The standard framework for sequential learning involves a tight feedback loop, where in each time-step an agent chooses an action, sees the corresponding reward, and can then update their beliefs. In a vaccine trial, however, we cannot immediately assess success after an innoculation; rather, we can only wait and see whether the patient gets infected any time before the end of the trial.

There has been some recent work on adaptive trials with delayed feedback [Grover et al., 2018, Joulani et al., 2013, Zhou et al., 2019]; however, available methods cannot simultaneously address some key difficulties that arise when testing vaccines in a pandemic. First, there is no (useful) upper bound on the delay separating an action and the corresponding reward. Study subjects could experience a negative reward (i.e., get infected) anytime between when they're enrolled in the study (and given a vaccine) and the end of the

---

[1]In this paper, we use the terms sequential learning and sequential experimentation interchangeably.

*Accepted for the 38th Conference on Uncertainty in Artificial Intelligence* (UAI 2022).

study. Second, the rate of infections doesn't just depend on which vaccines are used, but also on the ebb and flow of the pandemic. Any method that adaptively adjusts vaccine allocation frequencies without accounting for varying baseline infection rates risks providing a biased comparison. The main goal of this paper is to develop methods for sequential experimentation that can handle the above challenges.

Our core proposal is to extend the Thompson sampling method for sequential Bayesian learning to the proportional hazards model [Cox, 1972], which is widely used in medical statistics. In our context—as spelled out in more detail below—the proportional hazard model posits that, at time $t$ and for an as-of-yet uninfected person having received vaccine $k$, the instantaneous risk (i.e., hazard) of getting infected is of the form $h_0(t)e^{-\theta_k}$. Here, $h_0(t)$ is the baseline hazard, i.e., the time-varying instantaneous risk that an unvaccinated person gets infected, and $\theta_k$ captures the protective effect of the vaccine (the larger $\theta_k$ the better the vaccine). The proportional hazards model is a natural fit for our setting in that it allows us to address the challenges highlighted above (i.e., unbounded delays to observed infections and time-varying baseline hazards), yet it has enough structure to enable sample-efficient learning. One celebrated property of the proportional hazards model is that we can learn about the underlying efficiency parameters $\theta_k$ via a partial likelihood in which the baseline risk $h_0(\cdot)$ gets canceled out [Cox, 1975, Efron, 1977].

Our proposed approach, partial likelihood Thompson sampling (PLTS), involves running Thompson sampling with belief updates determined by partial likelihood each time we observe an event (i.e., each time an already vaccinated study participant gets infected). This differs from a Bernoulli bandit in that we do not control when events may happen, and the relevant "at risk" sample size changes with time. The resulting Bayesian problem doesn't have a closed-form solution for the posterior, but we find the setting to be amenable to popular methods for approximate inference with Thompson sampling—including Laplace approximation [Chapelle and Li, 2011, Russo et al., 2018]. While the use of partial likelihood for Bayesian inference in general is well established [Kalbfleisch, 1978], we are not aware of prior research on using proportional hazards modeling or partial likelihood for sequential experiments.

In a semi-synthetic study using data from the COVID-19 pandemic, we find that our approach can more reliably identify the best vaccine than a classical randomized controlled trial (RCT), in which volunteers are assigned to different treatments uniformly at random throughout the trial. Our approach also considerably reduces the within-experiment regret from assigning study participants to sub-optimal vaccines.

## 1.1 RELATED WORK

At a high level, sequential vaccine experiments can be seen as a bandit problem with partial, delayed feedback: Feedback is partial because we only ever observe negative rewards (infections), and delayed because it takes time for a study participant to potentially get infected post vaccination. There is a large amount of work on bandits with full feedback: Dudik et al. [2011] consider bandits with constant deterministic delays; Joulani et al. [2013] study a setting where delays have bounded expectation; Mandel et al. [2015] consider bounded delays in the stochastic multi-armed bandit problem; Thune et al. [2019] work with bounded delays in the nonstochastic bandit problem. Meanwhile, Vernade et al. [2017] allow for partial feedback but assume i.i.d. delays with a known distribution, and Gael et al. [2020] consider the same partially observable model but with the assumption that delay distributions satisfy polynomial tail bounds. Lancewicki et al. [2021] develop algorithms based on UCB and successive elimination that allow for unrestricted delay distributions. However, their bounds are vacuous in our scenario as we only observe infections (i.e., negative rewards); and the delays considered in Lancewicki et al. [2021] are assumed to be i.i.d across time. Thus, we are not aware of existing methods studied in a setting that includes vaccine trials, where delays are unbounded and time-varying, and positive rewards are never observed. We do note, however, that the method of Thune et al. [2019] is one that—at least algorithmically—could be plausibly considered in our setting, and we use it as a baseline in our experiments; see Section 4 for details.

Proportional hazards modeling and partial likelihood are core techniques in survival analysis. Cox [1972] first proposed the proportional hazards model, while Cox [1975] and Efron [1977] further developed statistical theory for estimators based on partial likelihood. Kalbfleisch and Prentice [1973], Breslow [1974], Efron [1977] provided alternative likelihood formulas when the event times are discrete with multiplicity. We also note a line of work justifying the use of Bayesian methods on partial likelihood. Kalbfleisch [1978] show that partial likelihood is a limiting marginal posterior under noninformative priors for baseline hazards. Sinha et al. [2003] further extend the result to scenarios with time-dependent covariates and time-varying regression parameters. Ibrahim et al. [2014] gives a comprehensive textbook treatment of Bayesian survival analysis.

Finally we note that using hazard rates to model the efficacy of vaccines is widely used in medical statistics; see for example Longini Jr and Halloran [1996], Durham et al. [1998] and Halloran et al. [1999]. Thus, the main contribution of this paper is to leverage fundamental concepts in survival analysis and classical vaccine RCTs, i.e., proportional hazards modeling and partial likelihood, to develop a new bandit algorithm suitable for adaptive vaccine trials.

## 2 INFECTION MODELING VIA PROPORTIONAL HAZARDS

We model vaccine trials as follows. At the start of the trial (i.e., at time $t = 0$), some participants are recruited to the trial and assigned to each vaccine group (arm) uniformly at random. After the initial assignment, volunteers arrive over time, and we randomize and assign them to a treatment arm as soon as they arrive. After enrollment, participants are followed until either they get infected or the study ends; any infected participants are removed from the study at the moment they are infected. Throughout, we use the following notation:

$$
\begin{aligned}
M_{t,k} &= \text{\# participants assigned to arm } k \text{ by time } t \\
m_{t,k} &= \text{\# participants assigned to arm } k \text{ at time } t \\
N_{t,k} &= \text{\# observed infections in arm } k \text{ by time } t \quad (1)\\
n_{t,k} &= \text{\# observed infections in arm } k \text{ at time } t \\
o_{t,k} &= \text{\# participants remaining in arm } k \text{ at time } t,
\end{aligned}
$$

i.e., $M_{t,k}$ and $N_{t,k}$ are cumulative sums of $m_{t,k}$ and $n_{t,k}$ respectively, and $o_{t,k} = M_{t,k} - N_{t,k}$. We also denote the sum of these statistics across all arms as $M_t, m_t, N_t, n_t, o_t$. We also have the assumption that $n_{0,k} = 0$ for all $k$ since we do not observe any infection at the start of the trial and $m_{T,k} = 0$ for all $k$ since we do not assign any new participants when we end the trial.

This general model is formalized in Protocol 1. One important case of this study design is the batched setting we consider in this paper, where there are a finite number of time points participants can join the study and infections can be recorded. Specifically, at $t = 0, ...., T$, we collect $m_t$ newly arrived participants and assign them to different groups and we also observe a vector of new infections $(n_{t,1}, ..., n_{t,K})$. At time $T$, we end the experiment. This is summarized in Protocol 2.

We model person-specific infection risk using the classical notion of a hazard rate, as follows. We assume that each of the $k = 1, \ldots, K$ treatment arms is characterized by a hazard rate $h_k(t)$, which captures the instantaneous risk that a person in study arm $k$ becomes infected at time $t$. Below, note that $o_{t,k}$ denotes the number of still uninfected participants in arm $k$ at time $t$, and $h_k(t)$ describes the expected fraction of these participants who will become infected in the next instant [Cox and Oakes, 1984].

**Assumption 1.** For each study arm $k = 1, \ldots, K$, there is a hazard rate $h_k(t)$ such that, for all $0 < t < T$,

$$
h_k(t) = \lim_{dt \downarrow 0} \frac{1}{dt} \frac{\mathbb{E}_t[N_{t+dt,\,k} - N_{t,\,k}]}{o_{t,k}}, \quad (2)
$$

where $\mathbb{E}_t$ denotes expectations conditionally on information available at time $t$.

The key flexibility of Assumption 1 is that it allows infection risk to ebb and flow over time: There may be some periods

---

**Protocol 1** General Vaccine Trial
**Input**: Length of experiment $T$, number of vaccines $K$
Assign $m_0$ participants uniformly at $t = 0$
**while** $t \leq T$ **do**
    **if** $m_t \neq 0$ **then**
        Assign $m_t$ participants to vaccine groups.
    **end if**
    Observe a vector of infections $(n_{t,1}, ..., n_{t,K})$ and end trial for the infected participants.
**end while**

---

**Protocol 2** Discrete Time Vaccine Trial
**Input**: Length of experiment $T$, number of vaccines $K$
**for** $t = 0, 1, ..., T$ **do**
    Assign $m_t$ participants to vaccine groups.
    Observe a vector of infections $(n_{t,1}, ..., n_{t,K})$ and end trial for the infected participants.
**end for**

---

where very few people from any study arms are getting infected, and others where infections are highly prevalent in some arms. However, Assumption 1 does impose non-trivial structure on the problem: For example, it implies that the length of time a patient has been in the study does not affect their risk of getting infected.

Given Assumption 1, Halloran et al. [1999] defines vaccine efficiency in terms of a ratio of hazard functions. Suppose that one of the study arms (without loss of generality the first arm $k = 1$) is a placebo that does not provide any protection against infection. Then the efficiency of the $k$-th vaccine depends on $h_k(t)/h_1(t)$.

**Definition 1.** *Under Assumption 1, for each non-placebo arm $k = 2, \ldots, K$, the vaccine efficiency is*

$$
VE_k(t) = 1 - \frac{h_k(t)}{h_1(t)}. \quad (3)
$$

Given our assumptions so far, the vaccine efficiency $\text{VE}_k(t)$ may vary with time, which creates some potential ambiguity in defining what the best vaccine is. Our next major assumption is that vaccine efficiency doesn't change with time, i.e., equivalently, that the hazard functions follow the proportional hazards model of Cox [1972].

**Assumption 2.** For each study arm $k = 2, \ldots, K$, there is an efficiency parameter $\theta_k$ such that

$$
h_k(t) = h_1(t)e^{-\theta_k}, \quad \text{VE}_k(t) = 1 - e^{-\theta_k}. \quad (4)
$$

Given Assumption 2, the main task of interest in assessing vaccines' efficiency is to estimate the efficiency parameter $\theta_k$: The bigger the $\theta_k$, the more effective the vaccine is. To illustrate this with a concrete example, in initial studies, Polack et al. [2020] reported that the Pfizer COVID-19

vaccine was 95% effective in preventing infection while Baden et al. [2021] reported that the Moderna COVID-19 vaccine was 94.1% effective. In the context of our model, both of these points estimates correspond to an efficiency parameter $\theta \approx 3$.

*Remark* 1. We set $\theta_1 = 0$ so the placebo arm has the same hazard rate as the baseline hazard.

*Remark* 2. Here, for simplicity, we assume constant efficiency of the vaccine. We make this assumption because we are modeling the infection against a particular variant within the time span of the trial. So, for example, in studying COVID vaccine efficiency against the omicron variant, we would only count omicron infections as events, while ignoring infections with other variants (and $h_0(t)$ would be essentially 0 early in the pandemic until omicron got prevalent). Given the usual time of such vaccine trials (on the order of months) we think it is reasonable to consider a constant $\theta_k$. However, for longer experiments, it may be necessary to extend the model to allow for waning efficiency. We leave extensions to non-constant efficiency to future work.

One major advantage of the proportional hazards model is that it enables a simple approach to learning the efficiency parameters $\theta_k$ via partial likelihood [Cox, 1972, 1975, Efron, 1977], as follows. Let us first suppose that the infection times (event times) of our participants in the trial are continuous as in Protocol 1 and recall at time $t$ the number of participants in each group is characterized by the vector $(o_{t,1}, ..., o_{t,K})$, i.e., this is the number of participants in each group who have joined the study, been assigned a treatment, and have not yet been infected. Then, the conditional probability that a person in vaccine group $j$ is infected given that there is an infection at time $t$ is (let $\theta_1 = 0$):

$$p_j(\theta_2, ..., \theta_K \mid o_{t,1}, ...., o_{t,K})$$
$$= \frac{h_0(t)e^{-\theta_j}}{\sum_{k=1}^{K} o_{t,k} h_0(t) e^{-\theta_k}} = \frac{e^{-\theta_j}}{\sum_{k=1}^{K} o_{t,k} e^{-\theta_k}}.$$

The unknown baseline hazard function cancels out because of the proportional hazards assumption. Now suppose we have $J$ events (infections) happening at time $t_1 < t_2 < \cdots < t_J$ and event $j$ happened to group $I_j$. We can then form the following partial likelihood,

$$\ell(\theta_1, ..., \theta_K) = \prod_{j=1}^{J} \frac{e^{-\theta_{I_j}}}{\sum_{k=1}^{K} o_{t_j,k} e^{-\theta_k}}, \tag{5}$$

which is a product over all the conditional probabilities of the observed events. It is a partial likelihood because we ignore all non-events. However, it is efficient for estimating the hazard rate parameters [Efron, 1977].

The partial likelihood (5) we obtained in the last section assumes continuous infections times where there are no ties in a single event time $t_j$. However, in Protocol 2 the event

times will be $1, ..., T$ and there could be multiple infections in a single vaccine group if the hazard rate is really high. Recall the definition of $(n_{t,1}, ..., n_{t,K})$ which denotes the number of infections happened in each vaccine group during the time interval $(t-1, t]$ and $n_t$ which denotes the sum of infections across all vaccine groups. In this case the exact likelihood proposed in Cox [1972] is the following

$$\ell(\theta_1, ..., \theta_K) = \prod_{t=1}^{T} \frac{e^{-\sum_{k=1}^{K} \theta_k n_{t,k}}}{\sum_{l \in R(n_t)} e^{-\theta(l)}} \tag{6}$$

where $R(n_t)$ is the set of all possible sets of $n_t$ participants from the risk set $(o_{t,1}, .., o_{t,K})$ and $\theta(l)$ is the sum of all the $\theta$ values of the individuals in set $l$. Due to its complicated form, Breslow [1974] suggests using the following approximation

$$\ell(\theta_1, ..., \theta_K) = \prod_{t=1}^{T} \prod_{k=1}^{K} \left( \frac{e^{-\theta_k}}{\sum_{i=1}^{K} o_{t,i} e^{-\theta_i}} \right)^{n_{t,k}} \tag{7}$$

of the exact partial likelihood (6), and this is what we do in our approach.[2]

## 3 PARTIAL LIKELIHOOD THOMPSON SAMPLING

In this section, we describe our proposed algorithm for sequential experimentation, which we call Partial Likelihood Thompson Sampling (PLTS). Thompson sampling [Thompson, 1933] is a Bayesian heuristic for sequential experiments that chooses the actions at each round according to the posterior probability that the action maximizes expected reward. This is usually implemented by sampling, where we sample an instance of environment from the posterior and take the action that maximizes the expected reward [Russo et al., 2018].

In our setting our model parameters are efficiency parameters $\theta_2, ...\theta_K$ (recall we assume that $\theta_1 = 0$, i.e., that the first arm is a placebo). At each round we will get a sample of $(\theta_2, .., \theta_K)$ from the posterior and assign our participants accordingly. We start with uninformative prior for all the parameters as they are potentially unconstrained [Gelman et al., 2013]. Then, following the blueprint of Thompson sampling (see Algorithm 3), we update the posterior each time we collect new data; and here, we do so using the partial likelihood introduced in the previous section. The use of partial likelihood for Bayesian posterior updates is further discussed in Kalbfleisch [1978].

---

[2]Other approximations have also been proposed in case of ties, notably that of Efron [1977]. Here, we use the Breslow approximation due to its simplicity and the fact that in our experiments there are only a few ties—and so our results are not particularly sensitive to the approximation method we use.

Given this setup, it now remains to derive an efficient posterior sampling method for assigning new participants as they arrive. Since we put an uninformative prior on all the parameters $\theta_2, ..., \theta_k$ the posterior at time $t$ given observed data $\mathcal{D}$ will be

$$\mathbb{P}_t(\theta_2, .., \theta_K \,|\, \mathcal{D}) = \frac{p(\mathcal{D} \,|\, \theta_1, ..., \theta_K)p(\theta_1, ..., \theta_K)}{p(\mathcal{D})}$$
$$\propto \ell(\theta_1, ..., \theta_K) \qquad (8)$$
$$= \prod_{l=1}^{t} \prod_{k=1}^{K} \left( \frac{e^{-\theta_k}}{\sum_{i=1}^{K} o_{l,i} e^{-\theta_i}} \right)^{n_{l,k}}.$$

Now, one difficulty in using (8) directly is that efficiently sampling from the posterior is non-trivial. For computational tractability, we thus use the popular idea of replacing the exact posterior with its Laplace approximation [Basu and Ghosh, 2020, Chapelle and Li, 2011, Gomez-Uribe, 2016, Russo et al., 2018]. The main idea is as follows. Writing $\mathbb{P}_t(\theta_2, .., \theta_K)$ for the partial likelihood used in (8), we see that ignoring constants,

$$\log \mathbb{P}_t(\theta_2, .., \theta_K \,|\, \mathcal{D})$$
$$= \sum_{l=1}^{t} \left( -\sum_{k=1}^{K} n_{l,k}\theta_k - n_l \log \left( \sum_{i=1}^{K} o_{l,i} e^{-\theta_i} \right) \right)$$
$$= -\sum_{k=2}^{K} \sum_{l=1}^{t} n_{l,k}\theta_k - \sum_{l=1}^{t} n_l \log \left( o_{l,1} + \sum_{i=2}^{K} o_{l,i} e^{-\theta_i} \right).$$

From the above formula we see that the logarithm of the likelihood is concave, hence there exists a unique global maximizer of the likelihood. Laplace approximation involves approximating the posterior with a Gaussian distribution centered at the posterior mode; the inverse covariance matrix will be $-\nabla^2 \log \mathbb{P}_t(\hat{\theta}_2, ..., \hat{\theta}_K \,|\, \mathcal{D})$, where $\hat{\theta}_2, ..., \hat{\theta}_K$ are the posterior mode. Formally, this gives us an approximate sampling distribution

$$\hat{\theta}_2, ...., \hat{\theta}_K = \operatorname{argmax} \mathbb{P}_t(\theta_2, ..., \theta_K \,|\, \mathcal{D})$$
$$\hat{\Sigma} = -\nabla^2 \log \mathbb{P}_t(\hat{\theta}_2, ..., \hat{\theta}_K \,|\, \mathcal{D}) \qquad (9)$$
$$\mathbb{P}_t \approx \mathcal{N}(\hat{\theta}_2, ...., \hat{\theta}_K; \hat{\Sigma})$$

We can efficiently solve the maximization problem in (9) using any smooth convex optimization algorithm (for example Newton's method). Here, the Hessian is readily available given the form of the likelihood, and so approximate sampling of the posterior is computationally efficient.

Now we can proceed to formulate our algorithms. Given the posterior sampling scheme described above, there are two popular ways of running Thompson Sampling. The canonical way [Thompson, 1933, Chapelle and Li, 2011] samples parameters from the current posterior distribution and takes the action that maximizes the expected reward, which aims at achieving low regret [Agrawal and Goyal,

2012, 2017, Kaufmann et al., 2012, Russo and Roy, 2016]. In our setting, we sample $\theta_2, .., \theta_K$ using (9) and choose the vaccine group with the maximum $\theta$. Algorithm 3 gives the details.

In our experiments, we also consider a PLTS-based adaptation of the top-two Thompson Sampling algorithm of Russo [2020]. This adaptation, where the second best action is selected in any given round with fixed probability $\beta$, targets the best arm identification problem. Algorithm 4 details the algorithm. The algorithm takes in a parameter $\beta$ which indicates the probability of sampling from the second best arm. We fix $\beta = 0.5$, which Russo [2020] suggests as a safe default choice.

---

**Algorithm 3** Partial Likelihood Thompson Sampling

**for** $t = 0, ..., T - 1$ **do**
    **if** no infection has happened to any arm **then**
        Assign new participants uniformly randomly.
    **else**
        Update posterior using (9).
        **for** each $m_t$ newly arrived participant **do**
            Sample $(\theta_2, ..., \theta_K)$ and set $\theta_1 = 0$.
            Assign participant to group $\arg\max_i \theta_i$.
        **end for**
    **end if**
**end for**

---

**Algorithm 4** Top Two Partial Likelihood Thompson Sampling ($\beta$)

**for** $t = 0, ..., T - 1$ **do**
    **if** no infection has happened to any arm **then**
        Assign new participants uniformly randomly.
    **else**
        Update posterior using (9).
        **for** each $m_t$ newly arrived participant **do**
            Sample $(\theta_2, ..., \theta_K)$ and set $\theta_1 = 0$.
            $I \leftarrow \arg\max_i \theta_i$.
            Sample $B \sim \text{Bernoulli}(\beta)$.
            **if** $B = 1$ **then**
                Assign participant to group $I$.
            **else**
                **repeat**
                    Sample $(\theta_2, ..., \theta_K)$ and set $\theta_1 = 0$.
                    $J \leftarrow \arg\max_i \theta_i$.
                **until** $J \neq I$
                Assign participant to group $J$.
            **end if**
        **end for**
    **end if**
**end for**

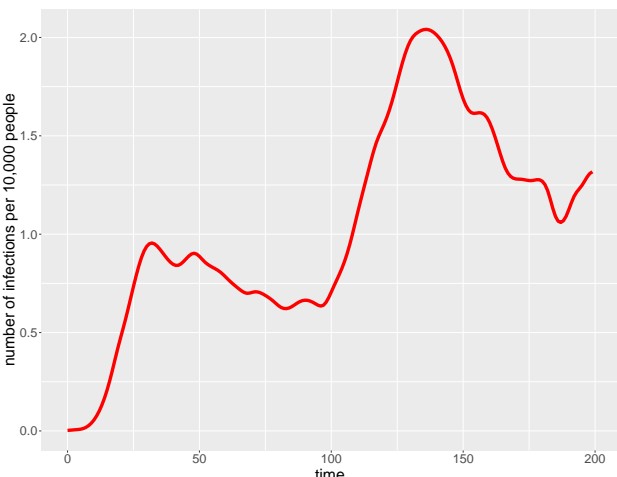

Figure 1: Baseline hazard rate as a function of time using moving average daily infections data provided by the CDC.

# 4 EVALUATION USING COVID-19 INFECTIONS DATA

We conduct semi-synthetic experiments in this section which is motivated by the real COVID-19 vaccines and case counts. Specifically, we model the baseline hazard rate using real world COVID-19 infections data and choose the efficiency parameters according to the efficacy of some approved vaccines. We evaluate our algorithm in both tasks, getting as few infections as possible and correctly identifying the best vaccine.

We simulate our experiment using Protocol 2 and fix our length of experiment to be $T = 200$. For simplicity we let the number of new participants to be constant at each time step, i.e. $m_t$ is a constant. Denote the total number of volunteers by $M$, we let $m_t = \frac{M}{T}$.

## 4.1 DATA

To model the baseline (or placebo) hazard rate, we use the data of 7-day moving average infections in US provided by the CDC data tracker [Centers for Disease Control and Prevention, 2021]. We pick the period of 200 days starting from March 9th, 2020. To get $h_1(t)$ we divide the daily infection numbers by the US population. The resulting baseline hazard rate is shown in Figure 1. We clearly see two distinct waves of infections that occurred during this 200-day period.

Our next task is to set the efficiency parameters $\theta_k$ corresponding to the non-placebo study arms. To do so, we use point estimates from a number of randomized controlled trials run early in the COVID-19 pandemic. Specifically:

- Based on AstraZeneca Vaccine trials with a 70% reported efficacy [Voysey et al., 2021], we set $\theta_2 = 1.2$.

- Based on SinoPharm Vaccine trials with a 78% reported efficacy [Al Kaabi et al., 2021], we set $\theta_3 = 1.5$.

- Based on Novavax Vaccine trials with an 89% reported efficacy [Heath et al., 2021], we set $\theta_4 = 2.2$.

- Based on Sputnik Vaccine trials with a 91% reported efficacy [Logunov et al., 2021], we set $\theta_5 = 2.4$.

- Based on Pfizer and Moderna Vaccine trials with roughly 95% reported efficacies [Polack et al., 2020, Baden et al., 2021], we set $\theta_6 = 3.0$.

The motivation for using these numbers is that we hope they capture realistic effect sizes one might see in a multi-arm vaccine trial, and not necessarily that they exactly match real-world efficiencies of the above vaccines established after pooling data from multiple trials.

## 4.2 EVALUATION METRICS

We evaluate the performance of each experimental design using the following metrics; throughout, we use the fact that the 6-th arm is best to condense notation.

- In-sample regret (ISR): Defined as $\frac{1}{T}\sum_{t=1}^{T} \theta_6 - \theta_{I_t}$ where $I_t$ is the action chosen at round $t$.

- Best arm identification probability (BIP), i.e., the fraction of times that the best arm (here, the 6-th) has the lowest estimated infection hazard. Specifically, let $A_i$ be the estimated best arm for replication $i$, the best arm identification probability is defined as $\sum_{i=1}^{B} \mathbf{1}\{A_i = 6\}/B$.

- Expected policy regret (EPR), as defined in Kasy and Sautmann [2021]: Let $a$ be the estimated best action, let $\Delta_a = \theta_6 - \theta_a$. This is defined as

$$\sum_{i=1}^{5} \frac{\sum_{j=1}^{B} \mathbf{1}\{A_j = i\}}{B} \Delta_a \qquad (10)$$

Of these metrics, the first measures the "cost" of running the experiment (i.e., how many study participants were assigned to suboptimal arms during the trial), while the latter two measure the quality of the findings from the study.

## 4.3 METHODS UNDER COMPARISON

Our goal is to evaluate our proposed method, PLTS, as well as the top-two Thompson sampling based variant designed for best-arm identification (TTPLTS). We compare these methods to two baselines: A standard, uniformly randomized controlled trial (RCT), and the delayed exponential weighting (DEW) algorithm of Thune et al. [2019].

As discussed in the related work section, we are not aware of any existing methods for sequential experimentation

**Algorithm 5** Delayed Exponential Weights (DEW)

> **Input:** Learning rate $\eta$, number of arms $K$
> Initialize weights $w_0^a = 1, \ \forall a = 1, ..., K$
> **for** $t = 1, .., T$ **do**
>     Let $p_t^a = \frac{w_t^a}{\sum_b w_t^b}$ for $a = 1, ..., K$
>     Place newly arrived volunteers according to distribution $\mathbf{p_t}$
>     Observe set of infections $(s, a)$ where $s$ is the time of enrollment and $a$ is the vaccine group
>     For each infection $(s, a)$, let $w_t^a = w_{t-1}^a \exp(-\eta \cdot \frac{1}{p_s^a})$
> **end for**

| | Metric | | |
|---|---|---|---|
| Method | ISR | BIP (%) | EPR |
| RCT | 385.08 (0.04) | 86.0 (1.1) | 0.090 (0.007) |
| $\eta = 0.01$ | 297.14 (7.71) | 86.6 (1.1) | 0.086 (0.007) |
| $\eta = 0.1$ | 186.03 (24.45) | 89.7 (1.0) | 0.067 (0.006) |
| $\eta = 0.4$ | 158.23 (39.80) | 79.3 (1.3) | 0.149 (0.010) |
| PLTS | 160.25 (0.96) | 91.8 (0.9) | 0.052 (0.006) |
| TTPLTS | 183.76 (0.71) | 93.5 (0.8) | 0.041 (0.005) |

Table 1: Results comparing Partial Likelihood Thompson sampling (PLTS), top-two PLTS (TTPLTS), DEW with varying learning rate ($\eta = 0.4, 0.1, 0.01$) and the randomized controlled trial (RCT). We display three metrics defined previously and fix $M = 60000$. Standard errors are given in parentheses; each configuration is replicated 1000 times.

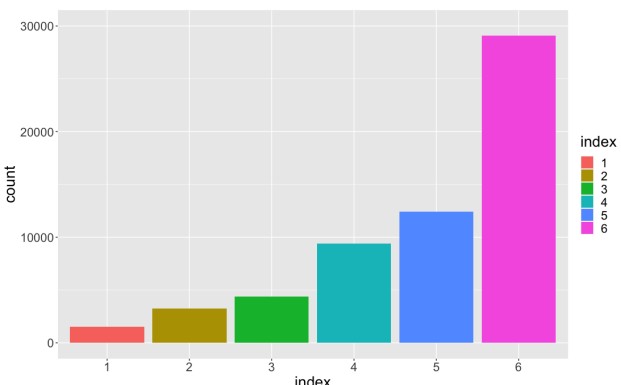

Figure 2: Total number of participants assigned to each vaccine group for the method PLTS with $M = 60000$ total participants, averaged over 1000 replications.

that were designed for our setting, i.e., with only negative feedback, unbounded delays in receiving feedback, and time-varying delays. However, the DEW approach, although introduced and studied in an adversarial setting with bounded delays, is simple and flexible enough that—at least algorithmically—it can be used in our setting, which is why we also explore using it as a baseline.

The DEW algorithm is a form of exponential weighting where weights are updated whenever negative rewards are observed; see Algorithm 5 for details. The one major challenge in using this algorithm is in choosing the learning rate $\eta$. Thune et al. [2019] offer guidance based on bounds on the delay distribution, but here of course we have no such bounds (and our setting does not fall under the purview of their theory), so it wasn't clear to us how to choose $\eta$. Thus, we simply consider 3 baselines, DEW with $\eta = 0.01, 0.1, 0.4$, that span the range of behaviors one can get from the method. (For Thompson sampling, we use an uninformative prior and so there is no analogous tuning parameter for the learning rate that needs to be specified.)

Finally, in order to evaluate best arm identification probabilities, we need each method to output a recommended best arm at the end of the experiments. PLTS and TTPLTS output the vaccine with the largest posterior mode and DEW outputs the vaccine with the largest weight. RCT picks the vaccine with lowest infection rate at the end of the trial.

## 4.4 RESULTS

For each method we consider, we use a sample size of $M = 60000$ study participants and replicate all simulations 1000 times. Table 1 summarizes the results across all methods and performance metrics.

Our first comparison is between the simplest variant of our methods, PLTS, and the RCT baseline. We here see that PLTS outperforms the RCT along all metrics: It both achieves smaller in-sample regret and has more power to identify the best arm. The reason it can do so is that it quickly shifts sampling towards the most promising vaccines; see Figure 2. This is clearly desirable from a regret minimization point of view, but here it is also desirable from a power point

of view since it concentrates sampling on the most difficult questions, i.e., distinguishing the best arms from each other.

Next, we compare PLTS to the DEW baselines in terms of in-sample regret. Here, the picture is nuanced. When well tuned, DEW can slightly outperform PLTS; however, it is not clear whether an adaptive tuning parameter choice could mirror this result. The rate at which all methods incur infections during the study is shown in Figure 3. We see that all the methods incur similar numbers of infections in the first wave, but the well-performing methods are able to focus on the better arms and considerably cut down on infections by the time we get to the second (larger) wave.[3]

The comparison looks different, however, once we look at metrics that consider the quality of the selected arm, i.e., best arm identification probability and policy regret. Here, PLTS still does well, but variants of DEW that achieved small

---

[3] One reason all study designs still have a fairly large number of infections in the second wave is that all designs assigned a non-trivial number of subjects to the less effective arms, including the placebo, at the beginning of the study, and that many of these participants were then vulnerable to infection in the second wave.

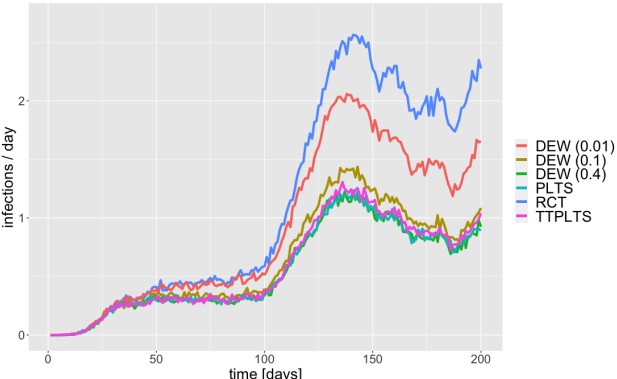

Figure 3: Total infections across all vaccine groups for 6 methods we consider: DEW with varying learning rate ($\eta = 0.4, 0.1, 0.01$), RCT, PLTS and TTPLTS, averaged over 1000 replications with total number of participants $M = 60000$.

in-sample regret do very poorly. It appears that, in order to achieve good in-sample regret, DEW needs to make unstable or greedy choices that hurt the quality of the selected arm. In contrast, PLTS is able to focus on the best arms without suffering from this phenomenon.

Relative to PLTS, the top-two variant TTPLTS achieves better post-trial metrics but worse in-sample regret. This is to be expected, since TTPLTS invests more in sampling the second-best arm in order to improve power for best arm identification. Whether a practitioner prefers the behavior of PLTS or TTPLTS will depend on the relative importance they give to in-sample versus post-trial performance metrics.

Finally, we investigate how arm-assignment probabilities of different methods evolve over time: Figure 4 shows the assignment probabilities averaged over 1000 replicates for each vaccine candidate as a function of time for both DEW and PLTS. The dashed horizontal line shows the uniform probability RCT uses. We see that in both cases the more promising candidates get larger shares as time goes on. However, we do see that when DEW uses a large learning rate (corresponding to the cases with good in-sample regret), the assignment probabilities almost flatten out as we approach the end of the trial, suggesting that by this point the learning rate has become too fast to enable reliable convergence to the best arm.

## 5 DISCUSSION

Sequential experiments have considerable potential to address challenges associated by new disease variants that emerge during a pandemic [Castillo et al., 2021]. However, the vaccine trial setting comes with a number of statistical challenges—including unbounded and time-varying delay distributions and partial feedback—that have not been considered in the context of existing bandit algorithms. We

introduced partial likelihood Thompson sampling, which adapts Thompson sampling to the setting of vaccine trials using fundamental modeling techniques that have been prevalent for decades in the survival analysis literature [Cox and Oakes, 1984]. We find our method to be a robust and performant option for sequential experimentation in an experiment built around data from the COVID-19 pandemic, thus highlighting its promise as a tool for quickly targeting the use of existing vaccines against a new disease variant. Additionally, our method can also be applied in other settings where the proportional hazards model is relevant; for example, in email marketing or online advertising. Our method deals with the complicated delay structure arising from these applications from a modeling perspective, thus opening the door for more efficient learning.

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

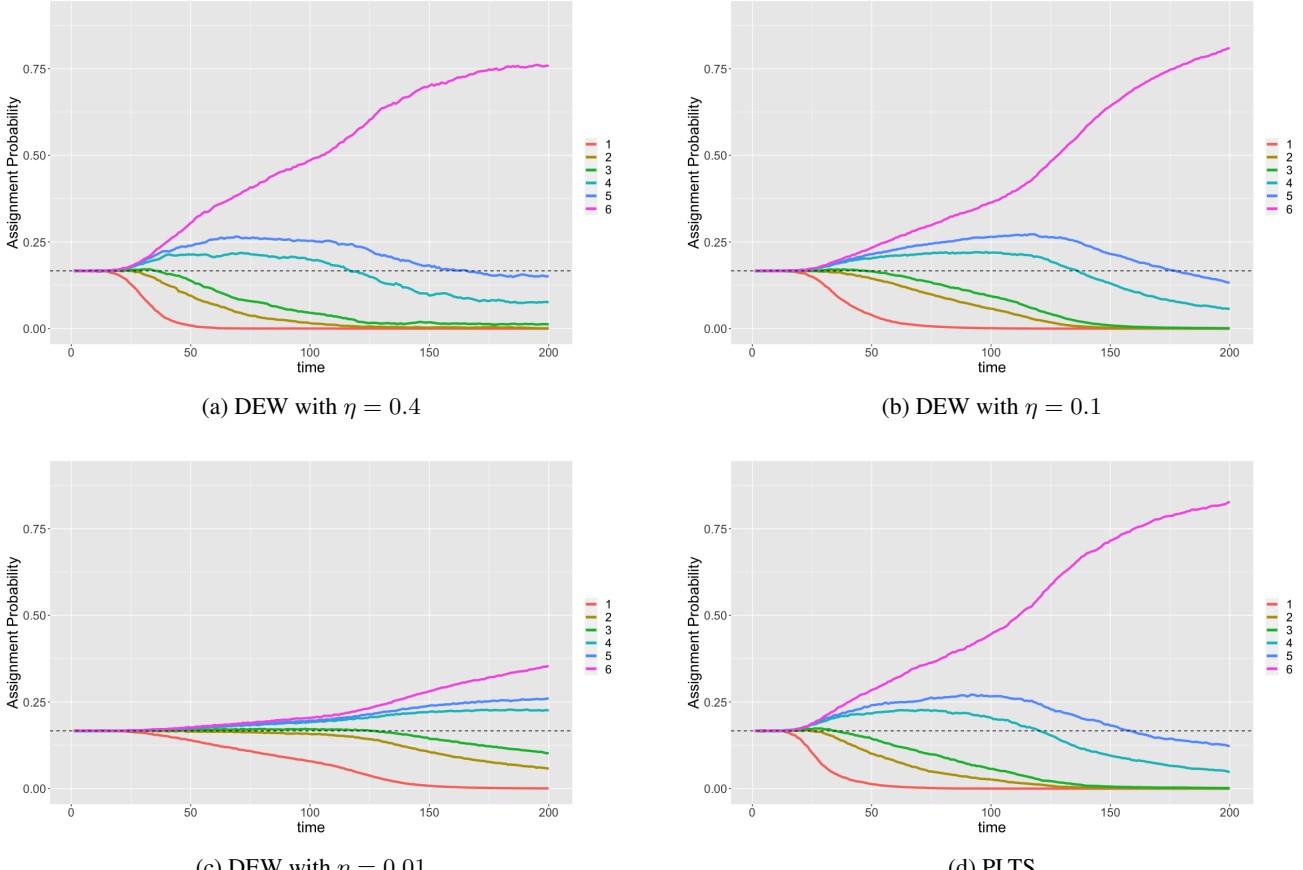

(a) DEW with $\eta = 0.4$

(b) DEW with $\eta = 0.1$

(c) DEW with $\eta = 0.01$

(d) PLTS

Figure 4: Plot of assignmnet probability of each vaccine group as a function of time for DEW with varying learning rate $\eta$ and PLTS when $M = 60000$, averaged over 1000 replicates.

Olivier Chapelle and Lihong Li. An Empirical Evaluation of Thompson Sampling. In *Advances in Neural Information Processing Systems*, volume 24, 2011.

David R Cox. Regression Models and Life-Tables. *Journal of the Royal Statistical Society: Series B (Methodological)*, 34(2):187–202, 1972.

David R Cox. Partial Likelihood. *Biometrika*, 62(2):269–276, 1975.

David R Cox and David Oakes. *Analysis of Survival Data*, volume 21. CRC Press, 1984.

Miroslav Dudik, Daniel Hsu, Satyen Kale, Nikos Karampatziakis, John Langford, Lev Reyzin, and Tong Zhang. Efficient Optimal Learning for Contextual Bandits. In *27th Conference on Uncertainty in Artificial Intelligence*, page 169–178, 2011.

L. Kathryn Durham, Jr. Longini, Ira M., M. Elizabeth Halloran, John D. Clemens, Nizam Azhar, and Malla Rao. Estimation of Vaccine Efficacy in the Presence of Waning: Application to Cholera Vaccines. *American Journal of Epidemiology*, 147(10):948–959, 05 1998.

Bradley Efron. The efficiency of Cox's likelihood function for censored data. *Journal of the American Statistical Association*, 72(359):557–565, 1977.

Kris Johnson Ferreira, David Simchi-Levi, and He Wang. Online Network Revenue Management using Thompson Sampling. *Operations research*, 66(6):1586–1602, 2018.

Manegueu Anne Gael, Claire Vernade, Alexandra Carpentier, and Michal Valko. Stochastic bandits with arm-dependent delays. In *37th International Conference on Machine Learning*, pages 3348–3356, 2020.

A. Gelman, J.B. Carlin, H.S. Stern, D.B. Dunson, A. Vehtari, and D.B. Rubin. *Bayesian Data Analysis, Third Edition*. Taylor & Francis, 2013.

Carlos Alberto Gomez-Uribe. Online Algorithms For Parameter Mean And Variance Estimation In Dynamic Regression Models, 2016.

Aditya Grover, Todor Markov, Peter Attia, Norman Jin, Nicolas Perkins, Bryan Cheong, Michael Chen, Zi Yang, Stephen Harris, William Chueh, and Stefano Ermon. Best arm identification in multi-armed bandits with delayed

feedback. In *21st International Conference on Artificial Intelligence and Statistics*, Proceedings of Machine Learning Research, pages 833–842. PMLR, 2018.

M. Elizabeth Halloran, Jr. Longini, Ira M., and Claudio J. Struchiner. Design and Interpretation of Vaccine Field Studies. *Epidemiologic Reviews*, 21(1):73–88, 03 1999.

Paul T. Heath et al. Safety and Efficacy of NVX-CoV2373 Covid-19 Vaccine. *New England Journal of Medicine*, 385(13):1172–1183, 2021.

Joseph G. Ibrahim, Ming-Hui Chen, and Debajyoti Sinha. *Bayesian Survival Analysis*. American Cancer Society, 2014. ISBN 9781118445112.

Pooria Joulani, Andras Gyorgy, and Csaba Szepesvari. Online learning under delayed feedback. In *30th International Conference on Machine Learning*, 2013.

J. D. Kalbfleisch and R. L. Prentice. Marginal Likelihoods Based on Cox's Regression and Life Model. *Biometrika*, 60(2):267–278, 1973.

John D Kalbfleisch. Non-Parametric Bayesian Analysis of Survival Time Data. *Journal of the Royal Statistical Society: Series B (Methodological)*, 40(2):214–221, 1978.

Maximilian Kasy and Anja Sautmann. Adaptive Treatment Assignment in Experiments for Policy Choice. *Econometrica*, 89(1):113–132, 2021.

Emilie Kaufmann, Nathaniel Korda, and Rémi Munos. Thompson Sampling: An Asymptotically Optimal Finite-Time Analysis. In *23rd International Conference on Algorithmic Learning Theory*, ALT'12, page 199–213, Berlin, Heidelberg, 2012. Springer-Verlag.

Talia Kustin et al. Evidence for increased breakthrough rates of SARS-CoV-2 variants of concern in BNT162b2-mRNA-vaccinated individuals. *Nature Medicine*, 27:1379–1384, 2021.

T.L Lai and Herbert Robbins. Asymptotically Efficient Adaptive Allocation Rules. *Advances in Applied Mathematics*, 6(1):4–22, 1985.

Tal Lancewicki, Shahar Segal, Tomer Koren, and Yishay Mansour. Stochastic Multi-Armed Bandits with Unrestricted Delay Distributions. In *38th International Conference on Machine Learning*, pages 5969–5978, 2021.

Benjamin Letham, Brian Karrer, Guilherme Ottoni, and Eytan Bakshy. Constrained Bayesian Optimization with Noisy Experiments. *Bayesian Analysis*, 14(2):495–519, 2019.

Denis Y Logunov et al. Safety and efficacy of an rad26 and rad5 vector-based heterologous prime-boost covid-19 vaccine: an interim analysis of a randomised controlled phase 3 trial in russia. *The Lancet*, 397(10275):671–681, 2021.

Ira M. Longini Jr and M. Elizabeth Halloran. A Frailty Mixture Model for Estimating Vaccine Efficacy. *Journal of the Royal Statistical Society: Series C (Applied Statistics)*, 45(2):165–173, 1996.

Travis Mandel, Yun-En Liu, Emma Brunskill, and Zoran Popovic. The queue method: Handling delay, heuristics, prior data, and evaluation in bandits. *AAAI Conference on Artificial Intelligence*, 29(1), Feb. 2015.

Fernando P. Polack et al. Safety and Efficacy of the BNT162b2 mRNA Covid-19 Vaccine. *New England Journal of Medicine*, 383(27):2603–2615, 2020.

Daniel Russo. Simple Bayesian Algorithms for Best-Arm Identification. *Operations Research*, 68(6):1625–1647, 2020.

Daniel Russo and Benjamin Van Roy. An Information-Theoretic Analysis of Thompson Sampling. *Journal of Machine Learning Research*, 17(68):1–30, 2016.

Daniel J Russo, Benjamin Van Roy, Abbas Kazerouni, Ian Osband, and Zheng Wen. A Tutorial on Thompson Sampling. *Foundations and Trends® in Machine Learning*, 11(1):1–96, 2018.

Debajyoti Sinha, Joseph G. Ibrahim, and Ming-Hui Chen. A Bayesian Justification of Cox's Partial Likelihood. *Biometrika*, 90(3):629–641, 09 2003.

William R Thompson. On the likelihood that one unknown probability exceeds another in view of the evidence of two samples. *Biometrika*, 25(3/4):285–294, 1933.

Tobias Sommer Thune, Nicolò Cesa-Bianchi, and Yevgeny Seldin. Nonstochastic Multiarmed Bandits with Unrestricted Delays. In *Advances in Neural Information Processing Systems*, 2019.

Claire Vernade, Olivier Cappé, and Vianney Perchet. Stochastic Bandit Models for Delayed Conversions. In Gal Elidan, Kristian Kersting, and Alexander T. Ihler, editors, *33rd Conference on Uncertainty in Artificial Intelligence*. AUAI Press, 2017.

Merryn Voysey et al. Single-dose administration and the influence of the timing of the booster dose on immunogenicity and efficacy of chadox1 ncov-19 (azd1222) vaccine: a pooled analysis of four randomised trials. *The Lancet*, 397(10277):881–891, 2021.

Zhengyuan Zhou, Renyuan Xu, and Jose Blanchet. Learning in Generalized Linear Contextual Bandits with Stochastic Delays. In *Advances in Neural Information Processing Systems*, volume 32, 2019.
