# OpenReview forum: "Partial Likelihood Thompson Sampling"
_auai.org/UAI/2022/Conference — UAI 2022 Poster_

### Official Review · Reviewer_6opz · 2022-04-12

**Q2(1) Originality/Novelty:** 3
**Q2(2) Significance/Impact:** 3
**Q2(3) Correctness/Technical Quality:** 3
**Q2(6) Clarity Of Writing:** 3
**Q6 Overall Score:** 7
**Q8 Confidence In Your Score:** 4

**Q1 Summary And Contributions:**

The paper extends Thompson sampling for setups that involve varying base rates and censoring, by combining them with proportional hazards model. The task is motivated by vaccine trials and the proposed practical method is evaluated on semi-synthetic vaccine trials data.

**Q2 Assessment Of The Paper:**

More detailed information regarding each of these aspects is given below:

**Q2(4) Quality Of Experiments (Optional):**

4: Excellent: The experimental evaluation is comprehensive and the results are compelling.

**Q2(5) Reproducibility:**

3: Good: Key resources (e.g., proofs, code, data) are available and key details (e.g., proofs, experimental setup) are sufficiently well-described for competent researchers to confidently reproduce the main results.

**Q3 Main Strengths:**

The task is well motivated and (I believe) more general than the vaccine example, and the proposed solution is technically sound and sufficiently easy to understand and implement. This implies potentially high impact for the field.

**Q4 Main Weakness:**

The paper relies very heavily on the vaccine trial terminology and uses it also throughout the methodological section, and hence looks a bit off as a UAI paper.

**Q5 Detailed Comments To The Authors:**

Thompson sampling is, despite its simplicity, clearly a valuable tool for sequential experimentation. This work extends Thompson sampling by relaxing its strict assumptions and the result achieves two important properties: It is clearly an extension in need and the solution is still easy to understand and use. I really like the fact that simple Laplace approximation is here used as the core technical element, since this results in nice and clean expressions compared to fancier approximations yet the choice is well motivated by the unimodality.

The vaccine trial example serves as excellent motivation for the work and would be sufficient in itself, but I believe many of the other sequential experimentation tasks in online advertising have similar characteristics. I would at least want to see a bit of a speculation on which of the properties might be important in other applications, but ideally I would like the writing to be re-worked so that the technical sections would rely less on the vaccine trial case. Right now the paper is a bit odd in terms of writing: It is almost exclusively about vaccine trials and as such would fit better to a more specialised forum (perhaps some bioinformatics journal), but the technical solution is appropriate for UAI. It would be better if the method was described in more general terms, but with a separate subsection that then maps the terminology for the vaccine trial case -- and ideally another subsection that would do the mapping for another example domain, e.g. online advertising. Evaluation on vaccine trials alone is fine.

I kind of understand the reasoning you had -- Section 2 is about infection modelling and hence uses vaccine trial terminology whereas Section 3 is about the method itself and uses more general terminology -- but I still feel that Section 2 would work better with more neutral terminology, especially as it becomes before Section 3. Now people building on your work will need to re-invent terms for 'non-placebo arm', 'vaccine efficiency', 'infected participant' etc and the literature becomes more scattered when people end up using different names for these.

**Q7 Justification For Your Score:**

Useful and elegant extension for a core method. The score is pulled down by the writing which is not optimal for UAI due to very heavy use of the application terminology in technical sections.

**Q9 Complying With Reviewing Instructions:**

1: Yes.

---

### Official Review · Reviewer_bW28 · 2022-04-13

**Q2(1) Originality/Novelty:** 2
**Q2(2) Significance/Impact:** 3
**Q2(3) Correctness/Technical Quality:** 4
**Q2(6) Clarity Of Writing:** 4
**Q6 Overall Score:** 7
**Q8 Confidence In Your Score:** 3

**Q1 Summary And Contributions:**

The work considers the problem of adaptive experimentation, motivated by vaccine trials, specifically with difficulties of negative rewards with unbounded delay. The approach is based on applying a variant of Thompson Sampling with a Laplace approximation. In a motivating experiment based on real-world COVID-19 vaccine trials, the efficacy of the method is presented.

**Q2 Assessment Of The Paper:**

More detailed information regarding each of these aspects is given below:

**Q2(4) Quality Of Experiments (Optional):**

3: Good: The experimental evaluation is adequate, and the results convincingly support the main claims.

**Q2(5) Reproducibility:**

2: Fair: Key resources (e.g., proofs, code, data) are unavailable but key details (e.g., proof sketches, experimental setup) are sufficiently well-described for an expert to confidently reproduce the main results.

**Q3 Main Strengths:**

The paper presents a method sequential experimentation, which combines several existing ideas for related settings. However the work seems to be novel in applying this to the problem setting outlined here, as far as the reviewer could asses. The ideas are presented clearly and correctly.

An interesting experiment has been set up, with multiple performance metrics, and the authors elaborate extensively on the results in the text. A second experiment, to establish the robustness of the method, but the included results are well presented, hence a score of 3 out of 4 here. Data sources have been attributed, but source code has not been included.

Overall the quality of the paper in terms of clearness of writing, presentation of the problem setting, theory and experimental results is very high.

**Q4 Main Weakness:**

The presented method seems to be a somewhat straightforward combination of several existing ideas. The experiment included is very nice, but a second experiment to assess robustness of the  PLTS method would be ideal.


**Q5 Detailed Comments To The Authors:**

In the introduction: the terms "sequential learning", "sequential experimentation", "sequential decision making" and "adaptive experiments" seem to be used interchangeably, and upon reading it was unclear to me if these are different concepts.

In the experiment, DEW tuned to a learning rate of 0.4 performed worse than RCT across the two metrics that indicate explorative performance, which is slightly surprising. Could you comment on that, was the tuning here very much off and how did you pick the learning rates?

Regarding reproducibility, will the source code be made available?

In the discussion, PLTS is labelled as a robust method, does that refer to the lack of tuning required to, say, DEW?

Minor comments / typos:
- Abstract: "..available method"
- Sec2: eqn (1) is clear but introduce "arm" beforehand or relabel as vaccine.
- Sec2: "We also have the convention" is it a convention or just by design?
- Sec2: Capitals mid sentence after colons (see also Sec4.4)
- Sec3: "We see that ignoring constant" sentence unfinished?

**Q7 Justification For Your Score:**

Overall the method is novel, theory seems correct and the method is convincing accross perforamnce metrics in semi-synthethic experiment that uses real-world COVID-19 data.


**Q9 Complying With Reviewing Instructions:**

1: Yes.

---

### Official Review · Reviewer_dWEL · 2022-04-13

**Q2(1) Originality/Novelty:** 3
**Q2(2) Significance/Impact:** 3
**Q2(3) Correctness/Technical Quality:** 3
**Q2(6) Clarity Of Writing:** 3
**Q6 Overall Score:** 6
**Q8 Confidence In Your Score:** 3

**Q1 Summary And Contributions:**

This paper proposes a Thompson sampling algorithm for the case of vaccine trials, where the result of pulling an arm (administering a specific vaccine) will only be known if that person gets infected, and only with a delay. The Cox proportional hazards model is used for this purpose. An experimental study using semi-synthetic data shows promising results.

**Q2 Assessment Of The Paper:**

More detailed information regarding each of these aspects is given below:

**Q2(4) Quality Of Experiments (Optional):**

2: Fair: The experimental evaluation is weak: important baselines are missing, or the results do not adequately support the main claims.

**Q2(5) Reproducibility:**

3: Good: Key resources (e.g., proofs, code, data) are available and key details (e.g., proofs, experimental setup) are sufficiently well-described for competent researchers to confidently reproduce the main results.

**Q3 Main Strengths:**

Clearly written paper that could have a good impact.

**Q4 Main Weakness:**

There are no theoretical performance guarantees, and the experimental section only considers a single fixed setting. It would be interesting to see how the results are affected by relevant aspects of the setting, e.g. the difference between the two best $\theta$'s.

**Q5 Detailed Comments To The Authors:**

Questions and larger comments:
1. For what stage(s) in the drug approval process in this method intended?
2. The abstract suggests that this method will be able to deal with the situation where new virus variants become prevalent during the course of the study, but assumption 2 rules out this possibility. Please clarify that in the abstract.
3. In section 2, at some point the baseline and placebo hazard functions play similar roles, but are introduced separately. This redundancy confused me. Could you state their relation explicitly? (I believe they are actually the same since you set $\theta_1=0$.) Alternatively, you could postpone introducing the placebo until the experimental section.
4. For section 3, can you give a reference for the claim that in this model, the uniform prior is uninformative?
5. In algorithms 3 & 4, does the condition "no infection has happened" mean $n_t=0$ or $N_t=0$? I think it's the latter (even though there posterior doesn't need to be updated if $n_t=0$). Please make this explicit.
6. In section 4.2, the expressions for ISR and EPR are always $\leq 0$, but they should be $\geq 0$.
7. Also in section 4.2: Is the probability appearing in (10) the same as in BIP (but now also for arms 1-5)? Please either align the notation or explain the difference.

Minor:
- page 1: "however, when we" - remove "when"
- page 2: "plausibly considered" -> "plausibly be considered"
- page 2: "time dependent covariates" -> "time-dependent covariates"
- equations (3) and (8) are missing a final "."
- page 3: "vaccines effectiveness" -> "vaccines' effectiveness"
- is vaccine efficiency the same as vaccine effectiveness?
- page 5: "ignoring constant" -> "ignoring constants," (incl the comma)
- page 5 first display: add "$| \mathcal{D}$" to the probability. Also in the middle equation of (9), and a bit above (9).
- page 5: "targets best" -> "targets the best"
- page 5: "suggets" -> "suggests"
- page 6: "Moderns Vaccine" - isn't that "Moderna"?
- footnote 2: "including the study" - I think that should be "placebo"?
- references: I noticed a few cases where capitals were missing in titles, add braces to avoid this.

**Q7 Justification For Your Score:**

Good contribution; the limitation of the experimental section to a single setting is my main concern.

**Q9 Complying With Reviewing Instructions:**

1: Yes.

---

### Official Review · Reviewer_PGnV · 2022-04-19

**Q2(1) Originality/Novelty:** 3
**Q2(2) Significance/Impact:** 2
**Q2(3) Correctness/Technical Quality:** 3
**Q2(6) Clarity Of Writing:** 4
**Q6 Overall Score:** 6
**Q8 Confidence In Your Score:** 3

**Q1 Summary And Contributions:**

The authors present a method called partial likelihood Thompson sampling, that can handle challenges such as delayed feedback in sequential experiments. Thus, their method is claimed to be more practical in deciding what vaccines might have protective effects against new COVID-19 variants.

**Q2 Assessment Of The Paper:**

More detailed information regarding each of these aspects is given below:

**Q2(4) Quality Of Experiments (Optional):**

3: Good: The experimental evaluation is adequate, and the results convincingly support the main claims.

**Q2(5) Reproducibility:**

2: Fair: Key resources (e.g., proofs, code, data) are unavailable but key details (e.g., proof sketches, experimental setup) are sufficiently well-described for an expert to confidently reproduce the main results.

**Q3 Main Strengths:**

- The paper is well-written.

- The idea of extending the Thompson sampling method for sequential Bayesian learning to the proportional hazards model is novel and interesting.

**Q4 Main Weakness:**

- Assumption 2 is rather very strong. If the authors expect this method to be used in clinical trials, then assuming  vaccine efficiency doesn’t change with time is unrealistic and in my opinion makes the method ineffective.

- Given the importance of clinical trials, the authors do not discuss any potential limitations of their approach.


**Q5 Detailed Comments To The Authors:**

- Does the number of trial arms affect the proposed procedure in estimating the efficiency parameters? If so, how?

- Often, sequential experiments suffer from missing data and measurement error. I was wondering if the authors have thought about how to handle missing data in their approach?

- One conclusion the authors make is that “semi-synthetic study using data from the COVID-19 pandemic, we find that our approach can more reliably identify the best vaccine than a classical randomized controlled trial (RCT)”. This is not my area of expertise, but I wonder why that is? Is this purely because the proposed method can deal with “delayed feedback” in a more effective way? A brief discussion on this would be useful to the readers.


Typos:

- Abstract, [method]: prevalence make available methods inapplicable
- page 2 [out]: could plausibly considered in our setting
- page 2 [the]: note that using hazard rates
- Page 5, [suggets]: which Russo [2020] suggests as a safe default choice.

**Q7 Justification For Your Score:**

The topic is important and the proposed approach is novel, but there is a lack of discussion on why the proposed method is better than the other and what the limitations are if one were to adapt this in a trial.

**Q9 Complying With Reviewing Instructions:**

1: Yes.

---

### Decision · Program_Chairs · 2022-05-15

**Decision:**

Accept (Poster)

**Comment:**

Meta Review: This paper considers vaccine trials, where the result of pulling an arm (administering a specific vaccine) will only be known if that person gets infected, and only with a delay. The approach is based on applying a variant of Thompson Sampling with a Laplace approximation. It is used in combination with the Cox proportional hazards model. The proposed practical method is evaluated on semi-synthetic vaccine trials data and shows promising results.

There is a consensus among reviewers that the paper is well-written. The idea of extending the Thompson sampling method for sequential Bayesian learning is based on a combination of several existing ideas, but the application of these ideas to the problem outlined here is novel and interesting. A reviewer suggests that the idea presented here is more general than the vaccine example.

The experimental section only considers a single fixed setting, but what is presented here is thorough and interesting, and the authors elaborate extensively on the results in the text. Reviewers comment that they would like to see more experiments; for example, a second experiment to assess the robustness of the PLTS method would be ideal. But the authors replied, "Given the space constraints, we chose to thoroughly present one experiment that’s as realistic as possible rather than many more stylized experiments. However, we note that the behavior described in the experiment we show is representative of other settings we tried out early on."

The source code for data analysis has not been included, but the authors promise to release them if accepted.

Potential limitations include: (1) Assumption 2 that vaccine efficiency doesn’t change with time is rather strong; (2) There are no theoretical performance guarantees (no response to this). There is also a question of whether UAI is an appropriate venue for this paper, as it relies very heavily on the vaccine trial terminology and uses it also throughout the methodological section.